# Rationale for a Combination Therapy with the STAT5 Inhibitor AC-4-130 and the MCL1 Inhibitor S63845 in the Treatment of FLT3-Mutated or TET2-Mutated Acute Myeloid Leukemia

**DOI:** 10.3390/ijms22158092

**Published:** 2021-07-28

**Authors:** Katja Seipel, Carolyn Graber, Laura Flückiger, Ulrike Bacher, Thomas Pabst

**Affiliations:** 1Department for Biomedical Research, University of Bern, 2008 Bern, Switzerland; Carolyn.Graber@students.unibe.ch (C.G.); Laura.Flueckiger1@students.unibe.ch (L.F.); 2Department of Hematology, University Hospital Bern, 3010 Bern, Switzerland; veraulrike.bacher@insel.ch; 3Department of Medical Oncology, University Hospital Bern, 3010 Bern, Switzerland

**Keywords:** acute myeloid leukemia (AML), hematological malignancies, FMS-like tyrosine kinase 3 (FLT3), signal transducer and activator of transcription 5 (STAT5), ten-eleven translocation-2 (TET2), tumor suppressor p53 (TP53), myeloid leukemia cell differentiation protein (MCL1)

## Abstract

The FMS-like tyrosine kinase 3 (*FLT3*) gene is mutated in one-third of patients with de novo acute myeloid leukemia (AML). Mutated FLT3 variants are constitutively active kinases signaling via AKT kinase, MAP kinases, and STAT5. FLT3 inhibitors have been approved for the treatment of *FLT3*-mutated AML. However, treatment response to FLT3 inhibitors may be short-lived, and resistance may emerge. Compounds targeting STAT5 may enhance and prolong effects of FLT3 inhibitors in this subset of patients with *FLT3*-mutated AML. Here STAT5-inhibitor AC-4-130, FLT3 inhibitor midostaurin (PKC412), BMI-1 inhibitor PTC596, MEK-inhibitor trametinib, MCL1-inhibitor S63845, and BCL-2 inhibitor venetoclax were assessed as single agents and in combination for their ability to induce apoptosis and cell death in leukemic cells grown in the absence or presence of bone marrow stroma. Synergistic effects on cell viability were detected in both *FLT3*-mutated and *FLT3*-wild-type AML cells treated with AC-4-130 in combination with the MCL1 inhibitor S63845. AML patient samples with a strong response to AC-4-130 and S63845 combination treatment were characterized by mutated *FLT3* or mutated *TET2* genes. Susceptibility of AML cells to AC-4-130, PTC596, trametinib, PKC412, and venetoclax was altered in the presence of HS-5 stroma. Only the MCL1 inhibitor S63845 induced cell death with equal efficacy in the absence or presence of bone marrow stroma. The combination of the STAT5-inhibitor AC-4-130 and the MCL1 inhibitor S63845 may be an effective treatment targeting *FLT3*-mutated or *TET2*-mutated AML.

## 1. Introduction

FMS-like tyrosine kinase 3 (FLT3) is an inducible growth factor receptor signaling via PI3K-PDK1-AKT and via RAS-RAF-MEK-ERK leading to cell growth and proliferation [1,2]. Mutations of the *FLT3* gene are detected in around a third of patients with de novo acute myeloid leukemia (AML). Mutated FLT3 variants are constitutively active kinases signaling via AKT and MAP kinases, and as a gain of function via STAT5 [3]. The FLT3 inhibitor midostaurin was approved together with intensive chemotherapy for first-line treatment of *FLT3*-mutated AML by the FDA in 2017, and was also authorized for use in the EU. However, treatment response to FLT3 inhibitors may be short-lived, and leukemia relapse is the major cause of treatment failure, as resistance may frequently emerge [4]. Moreover, the stromal microenvironment provides an escape route from FLT3 inhibitors through the GAS6-AXL-STAT5 axis [5,6].

*STAT5* refers to two highly related genes, *STAT5A* and *STAT5B*, which are both located on human chromosome 17 [7]. STAT5 proteins are not only activated by a wide variety of ligands that control proliferation, survival, and cell communication, but their dysregulation also facilitates tumor progression in various human cancers, particularly leukemia and lymphoma [8]. The STAT5 proteins are key downstream transcription factors in *FLT3*-mutated AML. STAT5 inhibition was reported to be a promising strategy for FLT3-ITD+ AML treatment [5]. In AML cells, STAT5 can be activated by FLT3-ITD, but also by activated cytokine receptors [3,7,9]. Myeloid cytokine receptors can be activated by granulocyte and macrophage colony-stimulating factors (G-CSF, GM-CSF, M-CSF), stem cell factor (SCF), thrombopoetin (THPO), and interleukins. These cytokines are secreted by bone marrow stroma cells to support the growth of normal hematopoietic stem and progenitor cells in the bone marrow. The same cytokines will also support the growth of leukemic stem and progenitor cells in the bone marrow. In normal cord blood cells, STAT5 phosphorylation can be efficiently induced by THPO, IL-3, and GM-CSF. SCF-induced STAT5 phosphorylation is largely restricted to the megakaryocyte–erythroid progenitor (MEP) compartment, while G-CSF, as well as IL-3 and GM-CSF, are most efficient in inducing STAT5 phosphorylation in the myeloid progenitor compartments [10]. Stromal cells can induce multidrug resistance in AML cells via upregulation of the PI3K/Akt signaling pathway, or via upregulation of STAT3 signaling [11,12].

Compounds directly targeting STAT5 canonical functions may inhibit dimerization, DNA binding, or transcriptional activity. The mechanisms of direct STAT5 inhibition include disruption of tyrosine phosphorylation, dimerization, nuclear translocation, and/or DNA binding. Targeting the SH2 domain was, therefore, the main focus for the design and identification of selective inhibitors [13]. AC-4-130 directly binds to the STAT5 SH2 domain and disrupts STAT5 activation, dimerization, nuclear translocation, and STAT5-dependent gene transcription [14]. AC-4-130 impairs the proliferation and clonogenic growth of human AML cell lines and primary FLT3-ITD+ AML patient cells in vitro and in vivo. Activated STAT5 can block apoptosis via induction of BCL-2 and MCL-1 proteins [15,16]. The BCL2 inhibitor venetoclax was approved by the FDA in 2018 in combination with azacitidine, decitabine, or cytarabine for the treatment of newly diagnosed acute myeloid leukemia (AML) in adults who are 75 years or older, or who have comorbidities that preclude use of intensive induction chemotherapy. The MCL1 inhibitor S63845 was effective in combination with the MEK inhibitor trametinib in hematological cells with elevated MCL1- and MEK-protein levels, independent of the mutational status of *FLT3* and *TP53* [17].

Here, we assessed the STAT5-inhibitor AC-4-130, FLT3 inhibitor midostaurin, BMI-1 inhibitor PTC596, MEK-inhibitor trametinib, MCL1-inhibitor S63845, and BLC2 inhibitor venetoclax as single agents and in combination for their ability to induce apoptosis and cell death in leukemic cells grown in the absence or presence of bone marrow stroma. AML cells represented all major morphologic and molecular subtypes with normal karyotype, including *FLT3* mutated or wild-type, *NPM1* mutated or wild-type, as well as *TP53* mutated or wild-type cells.

## 2. Results

### 2.1. Susceptibility of AML Cell Lines Grown in the Absence or Presence of HS-5 Stroma Cells to AC-4-130 and Venetoclax

To determine the sensitivity of AML cells to the STAT5 inhibitor AC-4-130, AML cells were subjected to in vitro cytotoxicity assays. AML cells were treated with the compound for 20 h in dose-escalation experiments before cell-viability assessment. Cell viability was also determined in AML cells grown in the presence of bone marrow stroma cells secreting granulocyte and macrophage colony-stimulating factors (G-CSF, GM-CSF, M-CSF), and other cytokines thereby inducing STAT signaling. The AML cell lines covered the major morphologic and molecular subtypes including, particularly, *FLT3* mutated or wild-type, *NPM1* mutated or wild-type, as well as *TP53* wild-type, mutated, hemizygous, and null cells (Table 1).

The susceptibility to AC-4-130 was elevated in TP53-mutated SKM-1 cells, with IC_50_ of 0.7 µM; intermediate in TP53 wild-type MOLM-13 and ML-2 cell lines, with IC_50_ values of 3 µM; reduced in the TP53-mutant MOLM-16 and TP53 wild-type cell line OCI-AML3, with IC_50_ values of 10 µM; and very low in the TP53 null HL-60 cells (Figure 1A). In the presence of stroma cytokines, the susceptibility to AC-4-130 was enhanced in SKM-1 cells, with IC_50_ at 0.5 µM; and reduced in MOLM-13 and ML-2, with IC_50_ at 4.4 µM and 10 µM, respectively (Figure 1B). The sensitivity of AML cell lines to BMI1 inhibitor PTC596, MEK-inhibitor trametinib, and MCL1-inhibitor S63845 was determined in previous studies [17,18]. To determine the sensitivity of the AML cell lines to the BCL2 inhibitor venetoclax, dose-escalation experiments were performed. The susceptibility to venetoclax was elevated in ML-2 and MOLM-13, with IC_50_ values of 0.3 and 0.8 µM, respectively; intermediate in HL-60; and very low in SKM-1, MOLM-16, and OCI-AML3 cells, with IC_50_ > 10 µM (Figure 1C). In the presence of stroma cytokines, the susceptibility to venetoclax was much enhanced in SKM-1 cells, with IC50 of 0.4 µM, in OCI-AML3 with IC_50_ of 1.2 µM, and reduced in ML-2 with IC_50_ of 4 µM (Figure 1D). In order to define the most effective treatment combinations, we focused on inhibitors expected to elicit synergistic effects in combination with AC-4-130 based on previous studies with BMI1, FLT3, MCL1- and MEK inhibitors [17,18,19,20], as well as the BCL2 inhibitor venetoclax, as indicated in Figure 2.

### 2.2. AC-4-130 Combination Treatment in AML Cell Lines

Cell viability was determined in AML cell lines treated with increasing dosages of single compounds and in combination treatments using the STAT5 inhibitor AC-4-130 and a variety of targeted therapies, including the BMI-1 inhibitor PTC596, the MCL1 inhibitor S63845, and the MEK inhibitor trametinib (Figure 3). Combination indexes were calculated according to Chou Talalay (Table 2). Some cell lines were additionally treated with AC-4-130 in combination with the FLT3-ITD inhibitor PKC-412 (midostaurin) or the BCL2 inhibitor venetoclax. Synergistic effects were present in *TP53* mutated cell line SKM-1 treated with AC-4-130 and PTC596, trametinib, S63845, or venetoclax (Figure 3A, Appendix A). For the FLT3-ITD cell line MOLM-13, response to combination treatments was detected with moderate synergy with AC-4-130 in combination with PKC-412, S63845, or venetoclax, as well as additive effects with AC-4-130 in combination with PTC596, and antagonistic effects in the combination of AC-4-130 with trametinib (Figure 3B, Appendix A). The FLT3 and TP53 wild-type cell line ML-2 response was moderately synergistic to AC-4-130 in combination with S63845, and mildly synergistic in combination with trametinib or venetoclax (Appendix A).

### 2.3. Changed Susceptibility of AML Cell Lines to Combination Treatment with AC-4-130 in the Presence of Bone Marrow Stroma

To investigate the elevated susceptibility of FLT3 wild-type SKM-1 compared to FLT3-mutated MOLM-13 cells to AC-4-130, cell viability was determined in AML cells grown in the presence of bone marrow stroma cells secreting granulocyte and macrophage colony-stimulating factors (G-CSF, GM-CSF, M-CSF), and other cytokines thereby inducing STAT signaling. The STAT5 inhibitor AC-4-130 was more effective in SKM-1 (Figure 3C, Appendix A), and less effective in MOLM-13 cells grown in the presence of HS-5 stroma (Figure 3D), indicating a moderate cell- and context-specific dependence of STAT5 signaling. The BMI1 inhibitor PTC596 was more effective in SKM-1 and less effective in MOLM-13 cells grown in the presence of HS-5 cells, indicating an elevated cell and context dependence of BMI1 signaling. The BCL2 inhibitor venetoclax was more effective in SKM-1 and less effective in ML-2 cells in the presence of HS-5 stroma (Appendix A), indicating a cell and context dependence of BCL2 function. In contrast, the MCL1 inhibitor S63845 had similar efficacy in SKM-1 and MOLM-13 cells independent of HS-5 presence. The elevated susceptibility of SKM-1 cells to AC-4-130 may be due to dominant signaling of cytokine receptors via STAT5 in this cell line. SKM-1 cells’ response to the BCL2 inhibitor was very low in the absence, but substantial in the presence of HS-5 stroma, indicating BCL2 induction by cytokine receptor signaling in this cell line. In MOLM-13 cells, STAT5 signaling is induced by FLT3-ITD. Here, the presence of stroma cytokines may have activated other STAT proteins and reduced the susceptibility to the STAT5 inhibitor. In contrast to the STAT5 inhibitor AC-4-130 and the BMI1 inhibitor PTC596, whose efficacies were cell- and context-dependent, the MEK1 inhibitor trametinib was generally less effective in the AML cells in the presence of HS-5 stroma (Figure 3). While susceptibility to the MEK inhibitor trametinib was reduced in SKM-1 cells, MOLM-13 cells were resistant to trametinib in the presence of stroma cytokines. MEK1 may activate STAT5 in SKM-1, but not in MOLM-13 cells. For the FLT3 inhibitor midostaurin (PKC412), the susceptibility of MOLM-13 cells was reduced in the presence of HS-5 stroma. Because of the altered susceptibilities, the effects of combination treatments on AML cells grown in the absence of HS-5 stroma were altered in AML cells grown in the presence of HS-5 stroma (Table 2).

### 2.4. Induction of Apoptosis and Cell Death in AML Cell Lines

The effects of treatment with AC-4-130 and PTC596, S63845, or trametinib alone and in combination on induction of apoptosis, cell-cycle arrest, and cell death were determined in cytometric assays (Figure 4A–F), while effects on the expression of STAT5 target genes *CDKN1A* and *BCL2* were determined by qRT-PCR (Figure 4G,H). Apoptosis and cell death were strongly induced in MOLM-13 cells treated with AC-4-130 and further enhanced in combination with the BMI1 inhibitor PTC596 (Figure 4A,B). Apoptosis and cell death were also induced in MOLM-13 cells with AC-4-130 in combination with the FLT3 inhibitor midostaurin (PKC412) or the BCL2 inhibitor venetoclax (Figure 4C,D). Induction of apoptosis and cell death were less pronounced in SKM-1 cells treated with AC-4-130 in combination with trametinib, S63845, or PTC596 (Figure 4E). Here, the induction of cell-cycle arrest was substantiated as G1 arrest in midostaurin-treated cells and G2 arrest in PTC596-treated cells (Figure 4F), and evident in the induction of *CDKN1A* gene expression and reduction of *BCL2* gene expression in SKM-1 cells treated with AC-4-130 and trametinib (Figure 4G,H).

### 2.5. AC-4-140 Combination Treatments in Leukemic Cells In Vitro

After initial studies in AML cell lines, the treatment combinations of AC-4-130 with PTC596, trametinib, or S63845 were applied to patient-derived hematological cells. Primary cells of a variety of hematological malignancies were included to determine the specificity of the treatment combinations. A total of 18 AML, 1 MDS, 2 multiple myeloma (MM), 2 B-ALL, and 1 CML, as well as PBMCs of 4 healthy donors (HD), were subjected to single-compound and combination treatments (Table 3). Treatment with the STAT5 inhibitor AC-4-130 (2 µM) on its own had minimal effects on cell viability (Appendix A). Treatment with the BMI inhibitor PTC596 on its own or in combination with AC-4-130 had minimal effects on cell viability (Appendix AA). The cytotoxic effects induced by trametinib (100 nM) treatment were significant in a few AML samples, but not further enhanced in combination with AC-4-130 (Appendix AB). Cytotoxic effects induced by treatment with the MCL1 inhibitor S63845 (100 nM) were significant in 12 AML, the MDS sample and 1 B-ALL, with further enhancement in combination with AC-4-130 (Appendix AC). Treatment with the FLT3 inhibitor midostaurin (PKC412) on its own or in combination with AC-4-130 had minimal effects on cell viability (Appendix AD). Treatment with the BCL2 inhibitor venetoclax on its own had significant effects on cell viability in half of the tested AML samples, with further enhancement in combination with AC-4-130 (Appendix AE).

Compared to the healthy donor cells, the tested patient samples split into two groups with respect to response to combination treatment with AC-4-130 and S63845 (Figure 5). Cell viability was reduced to 75% in healthy donor cells, to 80% in AML cells with normal response (NR), and to 40% in AML cells with strong response (SR). The AML samples with strong response to AC-4-130 and S63845 combination treatment comprised eight *FLT3* mutated, five *TET2* mutated AML, and one AML with *IDH2* and *DNMT3A* mutations. One of the secondary AMLs susceptible to the combination of AC-4-130 and S63845 carried a *TP53* mutation with a variant allele frequency of 5%, while two AMLs less susceptible to this combination treatment carried *TP53* mutations with a variant allele frequency of 50% and 92%. Of the two B-ALL samples, one was very susceptible to the combination of AC-4-130 and S63845, while the other one had a normal response. However, to determine whether this treatment combination may be effective in B-ALL, a wider array of ALL samples would have to be tested.

## 3. Discussion

As treatment response to FLT3 inhibitors may be short-lived, with leukemia relapse as the major cause of treatment failure, compounds targeting STAT5 may enhance and prolong effects of FLT3 inhibitors in this subset of patients with *FLT3*-mutated AML. To characterize the susceptibility of various AML cell lines to the STAT5 inhibitor AC-4-130, we performed a dose-escalation screening. Susceptibility to AC-4-130 varied in the tested AML cell lines with elevated susceptibility of the *TP53*-mutated secondary AML cell line SKM-1, intermediate susceptibility of the *TP53* wild-type cell lines MOLM-13 and ML-2, as well as reduced susceptibility of the *TP53* wild-type cell line OCI-AML3, the *TP53* mutated MOLM-16, and the *TP53* null cell line HL-60. The elevated susceptibility of the SKM-1 cell line was unexpected, as there is no mutated FLT3 activating STAT5 signaling in these cells, and the presence of mutated tumor suppressor TP53 is often associated with chemo-resistance. In view of the cell-viability studies in AML patient cells, the susceptibility of SKM-1 cells to AC-4-130 may be due to the presence of a *TET2* mutation in this cell line. To investigate the elevated susceptibility of *FLT3* wild-type SKM-1 compared to *FLT3*-mutated MOLM-13 cells to AC-4-130, cell viability was determined in AML cells grown in the presence of bone marrow stroma secreting granulocyte and macrophage colony-stimulating factors (G-CSF, GM-CSF, M-CSF), and other cytokines thereby inducing STAT5 signaling. In the presence of bone marrow stroma, AC-4-130 and the BMI1 inhibitor PTC596 were more effective in SKM-1 cells and less effective in MOLM-13 cells, indicating that the elevated susceptibility of SKM-1 cells to AC-4-130 and PTC596 may be due to dominant signaling of cytokine receptors via STAT5 and via PI3K/AKT in this cell line. In addition, co-culture with HS-5 stroma may induce the STAT5 target protein BCL2, as has been suggested for HL-60 cells [25]. In contrast to AC-4-130 and PTC596, the MEK1 inhibitor trametinib and the FLT3 inhibitor PKC412 (midostaurin) were less effective in the AML cells in the presence of HS-5 stroma. HS-5 stroma may induce resistance to FLT3 and MEK inhibitors via upregulation of the PI3K/AKT signaling pathway [11]. Similar effects have been described in c-kit mutant AML cells, where cytokines secreted by bone marrow stromal cells protect c-KIT mutant AML cells from c-KIT inhibitor-induced apoptosis [26]. The differential efficacies of targeted compounds in the context of the peripheral blood and bone marrow environment may be relevant for therapeutic success. Early relapse may arise when treatments are effective at eradicating leukemic cells in the peripheral blood but not in the bone marrow, where leukemic cells are sheltered. Thus, novel biological systems are currently established that enable the investigation of leukemia–stroma cross-talk and verification of novel therapies’ effectiveness under such bone marrow niche-mimicking conditions [27]. In our study, we employed a simple co-culture system with the HS-5 stroma cell line to grow AML cells in a bone-marrow niche environment. We discovered that several targeted compounds had reduced efficacy in AML cells grown on HS-5 stroma, while the MCL1 inhibitor S63845 induced cell death with equal efficacy in the absence or presence of bone marrow stroma, making it an excellent candidate to target leukemic stem cells in the bone marrow.

To determine which compounds may be effective in AML cells treated in combination with a STAT5 inhibitor, we focused on inhibitors expected to elicit synergistic cytotoxic effects in combination treatments based on previous studies with BMI1, FLT3, MCL1, and MEK inhibitors [17,18,19,20]. In the current study, we found synergistic cytotoxic effects in *FLT3* wild-type AML cell lines treated with combinations of the STAT5 inhibitor AC-4-130 and the MCL1 inhibitor S63845, and in the FLT3-ITD-positive MOLM-13 cells treated with combinations of the STAT5 inhibitor AC-4-130 and the FLT3 inhibitor midostaurin (PKC412). However, concurrent STAT5 and FLT3 inhibitor treatment was not effective in *FLT3*-mutated AML patient samples. There were also clearly antagonistic effects observed in MOLM-13 cells treated with a combination of STAT5 inhibitor AC-4-130 and the MEK inhibitor trametinib. Moreover, the effects of concurrent STAT5 and MEK inhibition were rather antagonistic in AML patient samples, independent of *FLT3* status. This antagonistic effect in the presence of STAT5 and MEK inhibition may result from induced signaling via PI3K and AKT, thereby preventing induction of apoptosis.

To validate the findings in a translational setting, we applied the STAT5 inhibitor AC-4-130 in combination with BMI1, FLT3, MCL1, MEK, and BCL2 inhibitors to AML patient samples. While treatments with combinations of the STAT5 inhibitor AC-4-130 with the FLT3 inhibitor midostaurin (PKC412), the BMI1 inhibitor PTC596 or the MEK inhibitor trametinib were rather ineffective, the combination of AC-4-130 and the MCL1 inhibitor S63845 led to a significant reduction in cell viability in a larger subset of AML. The susceptible AML patient samples were characterized by the presence of either mutated *FLT3* or mutated *TET2*, and one case of mutated *IDH2* and *DNMT3A* genes. The susceptibility of *FLT3*-mutated cells to the STAT5 and MCL1 inhibitor may be based on the gain of function of the mutated FLT3 activating STAT5 [3] and its downstream target MCL1. The susceptibility of *TET2* mutant cells to the STAT5 inhibitor may be related to a cooperativity of TET2 and STAT5 in the activation of downstream targets [28,29]. The susceptibility of *TET2* mutated cells to the MCL1 inhibitor may arise due to AKT-induced MCL1 over-activation in these cells, where AKT may no longer be regulated by mutated TET2 protein [22,23]. Leukemic IDH1 and IDH2 mutations disrupt TET2 function and impair hematopoietic differentiation [30,31]. As TET2 function is dependent on the metabolites produced by the IDH proteins [21,32], susceptibility to STAT5 and MCL1 inhibitors may be similar in AML cells with *IDH* mutations, as in *TET2* mutated cells. 

Susceptibility to the combination treatment of AC-4-130 and S63845 was significant in a secondary AML carrying a *TP53* mutation with a variant allele frequency (VAF) of 5%. Two AML samples less susceptible to the combination of AC-4-130 and S63845 carried *TP53* mutations at frequencies of 50% and 92%. *TP53* mutation variant allele frequency may be associated with clinical outcome of patients with myelodysplastic syndrome (MDS) and AML [33,34]. In newly diagnosed AML, the presence of a *TP53* mutation with VAF > 40% was independently associated with a significantly higher cumulative incidence of relapse and worse relapse-free and overall survival in patients treated with a cytarabine-based regimen.

It is important to note that there was a considerable effect of the MCL1 inhibitor S63845 in combination with the STAT5 inhibitor AC-4-130 on cell viability of normal PBMCs of healthy donors. This is in contrast to the marginal effect of the MCL1 inhibitor S63845 in combination with the MEK inhibitor trametinib on normal PBMCs previously described in [17], indicating that MCL1 function is of vital importance in normal PBMCs in the context of activated STAT5 signaling, but not in the context of activated MEK signaling. Such a considerable effect on normal monocytes may constitute a limitation in a treatment strategy using a combination of STAT5 and MCL1 inhibitor in a therapeutic context. The side effects of a STAT5 and MCL1 inhibitor combination therapy may be comparable to the side effects of the conventional standard induction therapy, including elevated risk of infection and increased bruising or bleeding. A possible advantage of a STAT5 and MCL1 inhibitor combination compared to the standard induction therapy may be the effective reduction of leukemic stem cells in the bone marrow.

Compared to previous studies, we now, for the first time, included the bone marrow stroma in our preclinical testing of AML cells. This is a new approach that introduces a new level of testing of potentially effective combination therapies. New insights may be gained by restating preclinical testing in AML cells grown in the presence of bone marrow stroma.

## 4. Materials and Methods

### 4.1. Patient Samples

Mononuclear cells of AML patients diagnosed and treated at the University Hospital, Bern, Switzerland, between 2005 and 2018 were included in this study. Informed consent from all patients was obtained according to the Declaration of Helsinki, and the studies were approved by decisions of the local ethics committee of Bern, Switzerland. Peripheral blood mononuclear cells (PBMCs) and bone marrow mononuclear cells (BMMCs) were collected at the time of diagnosis before initiation of treatment. The AML cells were analyzed at the central hematology laboratory of the University Hospital Bern according to state-of-the-art techniques [35]. Mutational screening for FLT3, NPM1, TP53, and conventional karyotype analysis of at least 20 metaphases were performed in all samples. In addition, all samples were analyzed by NGS sequencing of the myeloid panel genes.

### 4.2. Cell Lines and Cell Culture

Human AML cells lines OCI-AML3 (AML-M4, *FLT3*wt, *DNMT3A* R882C, *NPM1*mut, *TP53*wt), MOLM-13 (AML-M5, t(9;11), *FLT3-ITD*, *TP53*wt), MOLM-16 (AML-M0, *FLT3*wt, *TP53*mut), ML-2 (AML-M4, t(6;11), *FLT3*wt, *TP53*mut), and HL-60 (AML-M2, *FLT3*wt, TP53 null) were supplied by the Leibniz Institute DSMZ, German Collection of Micro-Organisms and Cell Cultures. AML cells were grown in RPMI 1640 media (SIGMA-ALDRICH, St. Louis, MO, USA) supplemented with 20% fetal bovine serum (FBS, Biochrom GmbH, Berlin, Germany) in a standard cell culture incubator at 37 °C with 5% CO_2_. The human bone marrow stroma cell line HS-5 was obtained from ATCC (ATCC^®^ CRL-11882™). HS-5 cells were grown in DMEM media (SIGMA-ALDRICH, St. Louis, MO, USA) supplemented with 10% fetal bovine serum (FBS, Biochrom GmbH, Berlin, Germany). The HS-5 cells secreted granulocyte colony-stimulating factor (G-CSF), granulocyte-macrophage-CSF (GM-CSF), macrophage-CSF (M-CSF), Kit ligand (KL), macrophage-inhibitory protein-1 alpha, interleukin-1 alpha (IL-1alpha), IL-1beta, IL-1RA, IL-6, IL-8, IL-11, and leukemia inhibitory factor (LIF). For the co-culture assays, HS-5 cells were plated on six-well plates on day 1. On day 2, Nunc 0.4 uM cell culture inserts (ThermoFisher, Roskilde, Denmark) were placed over the HS-5 feeder layer and AML cells were filled into the cell culture inserts. On day 3, AML cells were collected from the six-well inserts and replated on 96-well plates, before addition of compounds. Cytotoxicity assays were performed on day 4.

### 4.3. Cytotoxicity Assays

For assays with AML cell lines, cells were plated at a density of 5 × 10^5^/mL and treated with targeted compounds or conventional induction therapy. For assays with patient-derived mononuclear cells, the cells were cultured for 2 h prior to treatment. The STAT5 inhibitor AC-4-130 was purchased at Aobious, Inc. Gloucester, MA, USA. The BMI1 inhibitor PTC596, the FLT3 inhibitor midostaurin (PKC412), the MCL1 inhibitor S63845, and the MEK inhibitor trametinib were purchased at MedChemExpress (Monmouth Junction, NJ, USA). A stock solution of venetoclax was prepared by dissolving a venclexta tablet (Abbvie Inc., North Chicago, IL, USA) in DMSO. Cell viability was determined after 20 h of treatment using the MTT-based cell-proliferation kit I (Roche Diagnostics, Mannheim, Germany). This time point was selected because the cellular responses were effectual for the calculation of combination indexes after 20 h of treatment with two compounds in leukemic cells. For AML cell lines, four independent assays (biological replicates) with four measurements (technical replicates) per dosage were performed. For hematological patient samples, two independent assays with three technical replicates per dosage were performed. Statistical analysis was done on GraphPad Prism (GraphPad Software, San Diego, CA, USA). Data are depicted as XY graphs, box plots or scatter plots with mean values and SD. For the calculation of combination indexes, three dosages of AC-4-130 and two dosages of the other compounds were applied alone and in combination. Combination indexes were calculated on CompuSyn software (version 1.0; ComboSyn, Inc., Paramus, NJ, USA).

### 4.4. Measurement of mRNA Expression by qPCR

RNA was extracted from AML cells and quantified using qPCR. The RNA extraction kit was supplied by Macherey-Nagel, Düren, Germany. Reverse transcription was done with MMLV-RT (Promega, Madison, WI, USA). Real-time PCR was performed on the QuantStudio 7 Real-Time PCR Instrument using FastStart Universal master mix (Roche Diagnostics, Mannheim, Germany) and gene-specific probes (Cat# 4331182, Thermo Fisher Scientific, Waltham, MA, USA) Hs00355782_m1 (CDKN1A), Hs04986394_s1 (BCL2), and Hs02758991_g1 (GAPDH). Measurements for CDKN1A and BCL2 were normalized with GAPDH values (ddCt relative quantitation). Assays were performed in three or more independent experiments. Data are depicted in column bar graphs plotting mean with SD values.

### 4.5. Antibodies and Cytometry

Staining for apoptosis was done using annexinV-CF488A (Biotium Inc., Fremont, CA, USA) in annexinV buffer and Hoechst 33,342 (10 μg/mL) for 15 min at 37 °C, followed by several washes. Propidium iodide was added shortly before imaging on a NC-3000 imager (ChemoMetec, Allerod, Denmark). For cell-cycle analysis, cells were incubated in lysis buffer with DAPI (10 μg/mL) for 5 min at 37 °C and analyzed on NC-3000 imager.

## Figures and Tables

**Figure 1 ijms-22-08092-f001:**
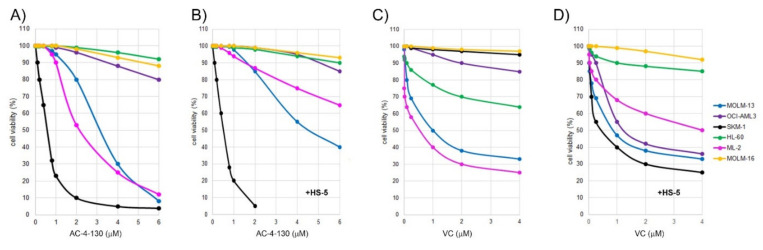
Dose-response curves of AML cell lines. AML cells were treated with the STAT5 inhibitor AC-4-130 (**A**,**B**) or the BCL2 inhibitor venetoclax (**C**,**D**) for 20 hours in the absence (**A**,**C**) or in the presence (**B**,**D**) of HS-5 stroma cells. Cell-viability data are average values of multiple repeat measurements per dosage. The standard deviation was 3–6%.

**Figure 2 ijms-22-08092-f002:**
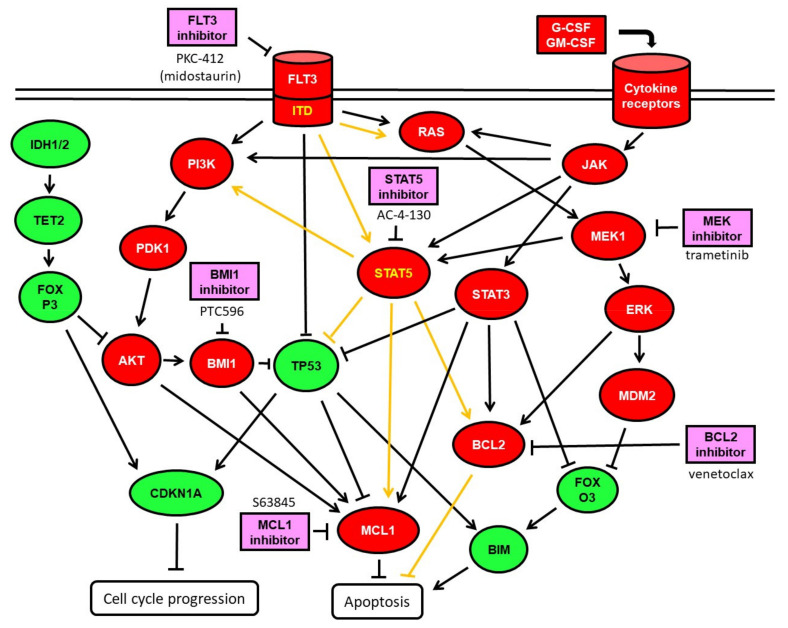
Schematic representation of STAT5 signaling pathways in myeloid cells. STAT5 can be activated by FLT3-ITD and by cytokine receptor signaling via Janus tyrosine kinases (JAKs). FLT3-ITD is a constitutively active growth factor receptor signaling via PI3K-AKT [2], via RAS-MEK-ERK [3], and via STAT5 [16], leading to cell growth and proliferation via p53 inhibition and MCL1 induction. Hematopoietic cytokine receptor signaling is largely mediated by JAKs and their downstream transcription factors, the STATs [7]. Mutations in the tumor suppressors IDH1/2 and TET2 may be functionally equivalent [21]. The tumor suppressor FoxP3 may inhibit AKT and its downstream target MCL-1 [22,23]. Oncogenic functions are indicated in red, tumor suppressor functions in green, and chemical inhibitors in pink.

**Figure 3 ijms-22-08092-f003:**
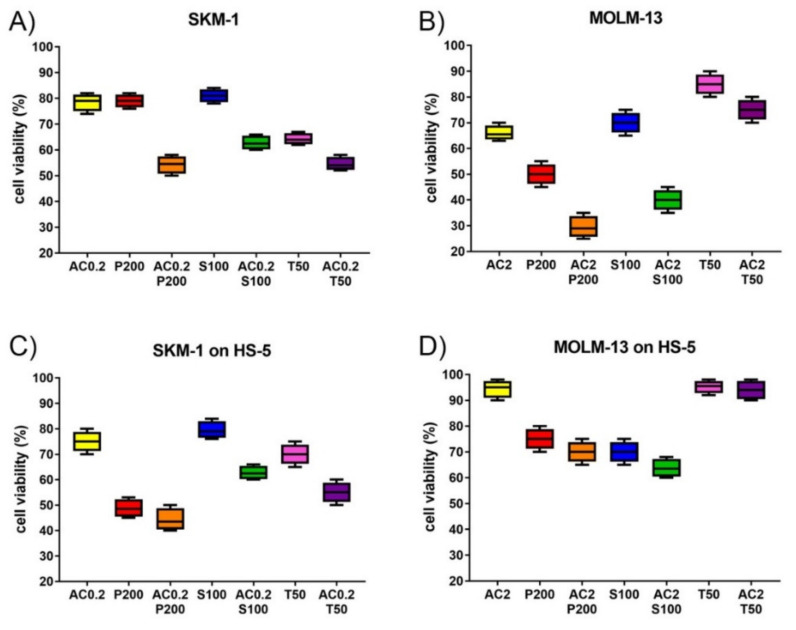
Susceptibility of SKM-1 and MOLM-13 AML cells to various treatment combinations. Cell viability was determined by AML cells grown in the absence (**A**,**B**) and in the presence (**C**,**D**) of HS-5 stroma cells. SKM-1 (**A**,**C**) and MOLM-13 (**B**,**D**) cells were treated for 20 hours with single compounds and in combination with AC-4-130 (AC, yellow) and PTC596 (P, red), S63845 (S, blue), or trametinib (T, pink). Concentrations of inhibitors are nM for PTC596, S63845, and trametinib, and µM for AC-4-130. All values are in reference to mock-treated cells (=100% viability).

**Figure 4 ijms-22-08092-f004:**
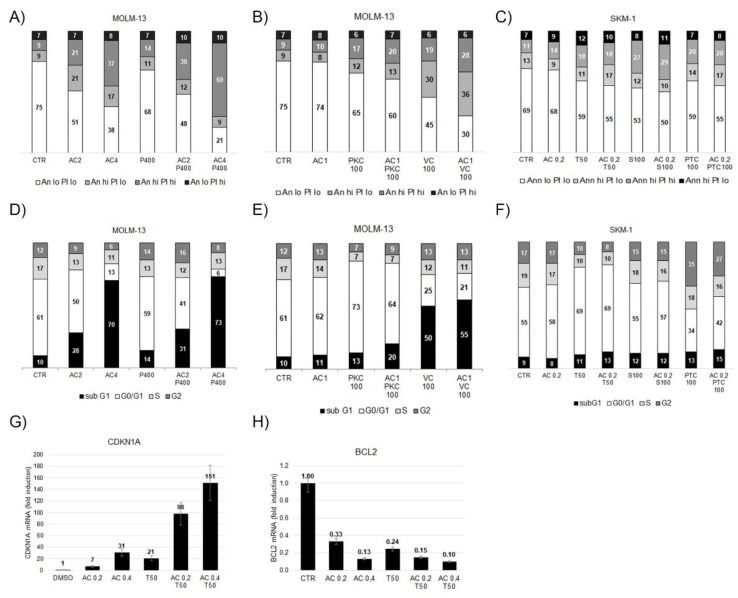
Induction of apoptosis and cell death in AML cells treated with AC-4-130 alone and in combination with targeted compounds. Cytometric analysis of MOLM-13 cells treated for 20 h with AC-4-130 alone and in combination with PTC596 (**A**,**D**), PKC412 or venetoclax (**B**,**E**) to measure induction of apoptosis using annexinV and PI staining (**A**,**B**), and induction of cell-cycle arrest and cell death (subG1 fraction) using DAPI staining (**B**,**E**). Cytometric analysis of SKM-1 cells treated for 20 h with AC-4-130 (AC) alone and in combination with trametinib (T), S63845 (S), or PTC596 to measure induction of apoptosis using annexinV and PI staining (**C**), and induction of cell-cycle arrest and cell death (subG1 fraction) using DAPI staining (**F**). Relative quantitation of STAT5A target genes *CDKN1A* and *BCL2* in SKM-1 cells treated for 20 hours with AC-4-130 and trametinib (**G**,**H**). Concentrations of inhibitors were nM for PKC412 (PKC), PTC596 (P), S63845 (S), trametinib (T), and venetoclax (VC), and uM for AC-4-130 (AC). Cells were grouped into four categories each: low- and high-intensity signal in Annexin staining (Ann lo, Ann hi), and low- and high-intensity signal in PI staining (PI lo, PI hi). DNA content in DAPI staining < 2n, 2n, >2n, 4n (subG1, G0/G1, S, G2).

**Figure 5 ijms-22-08092-f005:**
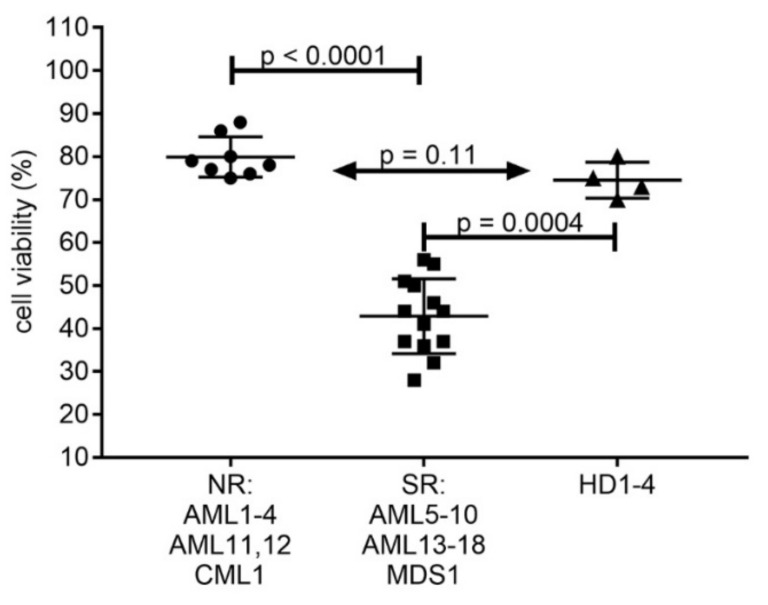
Susceptibility of hematological cells in vitro to combination treatment with AC-4-130 and S63845. Cell viability was determined in hematological cells after 20 h treatment with 2 µM AC-4-130 and 100 nM S63845The patient samples were sorted into two groups, one with normal response (NR) and one with strong response (SR). Significance was calculated by Mann-Whitney test. Primary data are presented in Appendix A.

**Table 1 ijms-22-08092-t001:** Genetic variants in acute myeloid leukemia (AML) cell lines.

ID	FAB	Origin	*FLT3*	*TP53*	Gene Variants	Karyotype
HL-60	M2	de novo	wt	null	*CDKN2A* R80X *NRAS* Q61L	hypotetraploid
ML-2	M4	de novo	wt	wt	*KMT2A*-AFDN *KRAS* A146T	t(6;11)
MOLM-13	M5a, relapse	MDS	ITD	wt	*KMT2A*-MLLT3	t(9;11)
MOLM-16	M0, relapse	de novo	wt	V173M C238S	*MLL* V1368L *MTOR* T571K	hypotetraploid
OCI-AML3	M4	de novo	wt	wt	*NRAS* Q61L *NPM1* L287fs *DNMT3A* R882C	+1, +5, +8
SKM-1	M5, refractory	MDS	wt	R248Q R248Q	*ASXL1* Y591X *KRAS* K117N *TET2* C1419fs	del9q12

AML, acute myeloid leukemia; MDS, myelodysplastic syndrome. FAB, French-American-British classification.

**Table 2 ijms-22-08092-t002:** Combination index values in AML cell lines.

AML Cell Line	HS-5 Stroma	AC-4-130 +PTC596	AC-4-130 +Trametinib	AC-4-130 +S63845	AC-4-130 +PKC412	AC-4-130 +Venetoclax
SKM-1	absent	0.6–0.8	0.6–0.8	0.7–0.9	nd	0.4–0.6
	present	1.0–1.2	0.8–1.0	1.1–1.3	nd	0.5–0.7
MOLM-13	absent	0.9–1.1	1.2–1.7	0.8–1.0	0.4–0.6	0.6–0.8
	present	0.6–0.8	nc	0.8–1.0	0.4–0.6	0.8–0.9

Combination indexes were calculated according to Chou Talalay [24]. Interpretation of combinatorial effects. Strong synergy CI = 0.1–0.3, moderate synergy CI = 0.3–0.7, mild synergy CI = 0.7–0.9, additive CI = 0.9–1.1, antagonism CI > 1.1. nc, not calculable; nd, not determined.

**Table 3 ijms-22-08092-t003:** Clinical characteristics of hematological patient samples.

ID	Disease	FAB	Antecedent	Gene Variants	Karyotype	AC+S *
AML1	sAML	M0	MDS	*EVI1* (overexpressed)	−7	NR
AML2	AML	M0		*TP53* (VAF 92%)	complex	NR
AML3	sAML		MDS	*CEPBA, ASXL1, EZH2, RUNX1*	normal	NR
AML4	sAML		ET	*ASXL1, CALR, KMT2A* (amp), *TP53* (VAF 50%),	49, +der(11)	NR
AML5	AML	M5		*FLT3-TKD* (0.63), *NPM1, DNMT3A*	normal	SR
AML6	sAML	M4	bicytopenia	*NPM1,**FLT3-TKD* (0.57), *TET2*, *TP53* (VAF 5%), *SRSF2*	+8	SR
AML7	AML	M5		*ASXL1*, *TET2*, *KRAS*, *SH2B3*, *U2AF1*	−7	SR
AML8	AML	M5		*FLT3-ITD* (0.58), *NPM1, DNMT3A*	normal	SR
AML9	sAML	M1	MDS	*NPM1*, *TET2*, *DNMT3A*	normal	SR
AML10	AML	M1		*FLT3-ITD* (0.833), *NPM1*	normal	SR
AML11	AML	M2		normal	complex, −7,−9	NR
AML12	sAML		MPN	*JAK2*	normal	NR
AML13	sAML	M4	breast cancer	*NPM1*, *TET2*	normal	SR
AML14	AML	M1		*FLT3-ITD* (>1.0), *NPM1*	normal	SR
AML15	AML	M4		*FLT3-ITD* (0.504), *NPM1*	normal	SR
AML16	AML	M2		*NPM1*, *IDH2*, *DNMT3A*	normal	SR
AML17	AML	M1		*FLT-3-ITD* (0.783), *BCOR*, *TET2*, *U2AF1*	del20q11, +8	SR
AML18	AML	M4		*FLT3-TKD* (0.487), *NRAS*, *KRAS*, *KMT2A-MLLT10*	t(10;11), +8	SR
MDS1	MDS			*TET2*, *ETV6*, *KRAS*, *SRSF2*, *CBL*	del7q	SR
BA1	B-ALL			IGH (rearranged)	normal	NR
BA2	B-ALL			normal	normal	SR
CML1	CML			*BCR-ABL1*	t(9;22)	NR
MM1	MM				normal	NR
MM2	MM				t(4;14)	NR
HD	n.a.			normal	normal	NR

AML, acute myeloid leukemia; sAML, secondary AML; ALL, acute lymphoblastic leukemia; CML, chronic myeloid leukemia; ET, essential thrombocytopenia; HD, healthy donor MDS, myelodysplastic syndrome; MPN, myeloproliferative neoplasm; MM, multiple myeloma. For the *FLT3* gene, the mutant allele ratio, and for the *TP53* gene, the mutant allele frequency (VAF) are indicated in parentheses. * Response to combination treatment with AC-4-130 and S63845; normal response (NR), strong response (SR).

## Data Availability

Data is contained within the article or the Appendix A.

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
