# Peer review of "Rationale for a Combination Therapy with the STAT5 Inhibitor AC-4-130 and the MCL1 Inhibitor S63845 in the Treatment of FLT3-Mutated or TET2-Mutated Acute Myeloid Leukemia"

_ijms, 2021, doi:10.3390/ijms22158092_

Round 1
Reviewer 1 Report
This is a very well performed and written study that analyzes the anti-leukemic effects of the STAT5-inhibitor AC-4-130 as mono- and combination therapy in a series of AML cell lines and primary patient specimens. Finally, the authors show that the combination of AC-4-130 and the MCL1 inhibitor S63845 might provide a reasonable approach for treating AML.
Although the interpretation is limited by the fact that in-vivo assays are missing, the manuscript creates interesting and valid data for follow-up projects.
I have only one comment/question/suggestion:
Did the authors try to perform Immunoblots for pSTAT5 and STAT5 in the AC-4-130 treated cells? Did the antileukemic effects correlate with inhibition of STAT5 phosphorylation? And were there differences in pSTAT5 the cells grown in the absence or presence of bone marrow stroma ?
Response: We thank the reviewer for this suggestion. It would be interesting to determine the impact of bone marrow stroma on STAT5 inhibition in different AML cells. We have shown that the STAT5 inhibitor AC-4-130 was more effective in SKM-1 and less effective in MOLM13 cells in the presence of HS-5 stroma. The expectation would be that these effects correlate with STAT5 phosphorylation. As we focused our attention to the expansion of the AML patient cohort tested in this study in order to define biomarkers of response, any studies on a correlation of STAT5 phosphorylation and anti-leukemic effects will have to be postponed.
We have added the following text to the introduction line 59:
Analysis of the subcellular localization of pY-STAT5 and STAT5 upon AC-4–130 treatment revealed reduced pY-STAT5 levels both in the cytoplasm and nucleus, as well as reduced overall levels of nuclear STAT5 in Ba/F3 FLT3-ITD+ cells in the absence of stroma (Wingelhofer et al., 2018). STAT5 phosphorylation may correlate to AC-4-130 susceptibility
and bone marrow stroma cytokines may affect the STAT5 phosphorylation levels in AML cells.
Reviewer 2 Report
In the manuscript “Rationale for a combination therapy consisting of the STAT5 inhibitor AC-4-130 in combination with the MCL1 inhibitor S63845 in the treatment of acute myeloid leukemia”, Katja Seipel et al. assessed the efficacy of the STAT5-inhibitor AC-4-130, the FLT3 inhibitor midostaurin (PKC412), the BMI-1 inhibitor PTC596, the MEK-inhibitor trametinib, the MCL1-inhibitor S63845, and the BCL-2 inhibitor venetoclax as single agents and in combination in acute myeloid leukemia (AML) cell lines in the absence or presence of bone marrow stromal cell lines. Furthermore, they tested the drugs in patient-derived AML cells and healthy donors’ PBMCs. The results suggested that the combination of the STAT5-inhibitor AC-4-130 and the MCL1 inhibitor S63845 may be an effective treatment targeting adverse risk leukemia including FLT3-ITD AML and TP53 mutated MDS and AML.
Response: We are very grateful for this thorough and detailed review. We feel that we have
incorporated all the raised suggestions thereby gaining relevant insights and improving the
quality and integrity of the manuscript to a considerable degree. In consequence we also
added the new conclusions to the title and abstract and amended our working model depicted
in Figure 2.
Importance in its field
Mutations in the MS-like tyrosine kinase 3 (FLT3) gene occur in almost one third of AML, in association with high leulemic burden and a poor prognosis. Targeted FLT3 signaling inhibitors have been identified, including midostaurin and gilteritinib, which are already approved by the FDA. Nevertheless, the responses to FLT3 inhibitor treatment are transient, explaining the need to investigate novel therapeutic strategies to address relapses and to prolong patient survival.
Seipel et al. already published work on combination therapies in AML (“BMI1-Inhibitor PTC596 in Combination with MCL1 Inhibitor S63845 or MEK Inhibitor Trametinib in the Treatment of Acute Leukemia”, 2021, “Rationale for a Combination Therapy Consisting of MCL1- and MEK-Inhibitors in Acute Myeloid Leukemiq”, 2019 and “MDM2- and FLT3-inhibitors in the treatment of FLT3-ITD acute myeloid leukemia, specificity and efficacy of NVP-HDM201 and midostaurin, 2018). These papers include substances which are also included in the current manuscript, but all propose a different treatment strategy and, therefore, somewhat contradict each other. To be able to come to a reasonable conclusion on the most promising of the suggested strategies, it would have been more conclusive to summarize and to compare all substances in one analysis or discuss in much more critical ways, why the new combination is supposed to be superior to the ones promoted previously.
Response: Scientific research is a progressive process. Our previous studies were published when we had sufficient data and novel insights to warrant publication. Compared to previous studies we now, for the first time, included the bone marrow stroma in our preclinical testing of AML cells. This is a new approach adding valuable new insights into preclinical testing of novel therapies. The new combination treatment with STAT5 and MCL1 inhibitor is not generally superior to the ones previously promoted, however, it may be a valid treatment option in specific patient subsets., e.g. FLT3mutated or TET2 mutated AML. To emphasize the importance of this new approach we have added the following text to the discussion:
Compared to previous studies we now, for the first time, included the bone marrow stroma in our preclinical testing of AML cells. This is a new approach introducing a new level of testing of potentially effective combination therapies targeting AML cells in both peripheral blood and bone marrow. New insights may be gained on therapeutic efficacies by restating previous preclinical studies in AML cells in the presence of bone marrow stroma.
Major concerns
- Consistency: In the introduction and throughout the discussion, the authors have a hypothesis that STAT5 inhibition may enhance/prolong treatment efficacy of FLT3 inhibition in FLT3 mutated AML. Please further explain this assumption and relate to your following statements on the combination of STAT5 inhibition and MCL1 inhibition. FLT3 inhibitors are only included in one experiment with the MOLM-13 cell line in Figure 4.C and D, which does not comply with it being the main focus of this work and the FLT3 mutation being one of the main evaluation criteria, especially since the results did not show significant correlations between response to the combination of STAT5 and MCL1 inhibitors and the FLT3 mutation status of each cell line or AML patient. In general, most assertions miss an explanation. What exact reasons do you suggest for the synergism of this combination?
Response: We have included more information on the rationale for testing STAT5 inhibitor, as well as BCL2, MCL1 and MEK inhibitors in FLT3 mutated AML in the introduction. The combination of midostaurin and AC-4-130 was effective in FLT3 mutated MOLM-13 cells and AML patient samples in the absence of bone marrow stroma, much less effective in the presence of stroma. The combination of STAT5 and MCL1 inhibitor, however, was effective in FLT3 mutant MOLM-13 cells and AML patient samples, and in FLT3 wildtype, TET2 mutant SKM-1 cells and AML patient samples.
- Significance: In the conclusions, the authors note that the proposed combination showed cytotoxic effects on PBMCs of healthy donors as well. This questions the assumption that this treatment strategy is AML specific and this study therefore presents valuable results for clinical application. Additionally, in Figure 5.D, the combination showed higher efficacy on primary B-ALL cells as compared to AML patients. Please consider evaluating if there is a significant benefit to AML patients, for example by summarizing all AML patients and all healthy donors in one blot and determine the statistical value of the effect of the MCL1 inhibitor in combination with the STAT5 inhibitor on cell viability. Furthermore, in the discussion (line 287f.), the authors postulate a beneficial effect of this treatment combination in patients with TP53 mutations, which they justify by the response of one AML patient. In Figure 5, it is noticeable that one other patient who also carries the TP53 mutation, did not respond well to this combination. Please reevaluate to verify your statement and to avoid overinterpretations.
Response: We have added several more AML patient samples and one additional healthy donor sample and included a grouped analysis of response to AC-4-130 and S63845 in the new Figure 5. The patient samples can be sorted into two groups, one with normal response (NR) and one with strong response (SR). All the FLT3 mutated or TET2 mutated AML samples are in the SR group. Two TP53 mutated samples with elevated VAF are in the NR group. Only one TP53 mutated sample with low VAF is in the SR group. We have removed any interpretation of potential treatment efficacy for AC4-130 and S63845 in TP53 mutated samples.
- Cohort size and composition: The authors performed experiments on AML cell lines as well as on primary AML cells in vitro. Please provide information on how you interpret and connect the results of both. Since the results of inhibitors in cell lines are not always applicable to primary cells, the use of primary AML cells derived from AML patients is more important and more conclusive. The authors present a cohort of 8 AML patients, which is very small and complicates the attempt to interpret these findings in consideration of statistical significance, especially since the response highly varies between these 8 patients. The authors also included primary cells of different entities, including myelodysplastic syndrome, multiple myeloma, chronic myeloid leukemia, and acute lymphocytic leukemia. Please explain your selection of different entities, and why you included them in the first place. Especially the high response of the B-ALL2 sample in Figure 5.D does not support the initial hypothesis of STAT5 inhibition in combination with MCL1 inhibition as a treatment strategy in AML, but rather speaks for an entity-unspecific response. Please consider a larger and more uniform cohort of AML patient samples, possibly as well as mouse model experiments to verify your statement in vivo, and the inclusion of clinical datasets, since the AML patients used in Figure 5 seem to be dividable in responders and non-responders, and correlations might be detectable to explain this division.
Response: We included primary cells of a variety of hematological malignancies to determine the specificity of the treatment combinations. Reduction of cell viability was similar in the CML, MM and healthy donor cells. Of the two B-ALL samples, one was very susceptible to the combination of AC-4-130 and S63845, the other one had normal response. This may indicate an entity-unspecific response. However, to determine whether this treatment combination may be effective in B-ALL, a wider array of ALL samples would have to be tested. We have analyzed more AML patient
samples and one additional healthy donor sample and included a grouped analysis of response to AC-4-130 and S63845 in the new Figure 5. We have included the clinical data and describe a potential correlation of response and genotype. The patient samples can be sorted into two groups, one with normal response (NR) and one with
strong response (SR). All the FLT3 mutated or TET2 mutated AML samples are in the SR group. We have discussed possible mechanisms of interaction of STAT5 and TET2. Further testing of larger cohorts in vitro and testing in a murine model may validate the proclaimed combination therapy in the future.
- Sources: Please be more detailed when reporting results from previous publications and be careful to name the source (line 46ff., line 69ff., Figure 2).
Response: We have added the missing references in the introduction and Figure 2
legend.
- Figures: Please consider adjusting your figures to be easier to interpret. Many of them are crowded, different entities or compounds are not differentiable (especially Figure 3, 4, and 5). Measures to improve the informative value may be for example:
- Different colors for different compounds/entities
- Only show relevant results, e.g. only one concentration in Figure 3, remove Figure 4.A (similar data already shown in Figure 3.B), remove conditions which are shown twice in Figure 4.A/C
- Include spaces between the bars of different compounds or compound combinations in Figure 4
- Figure 5.B-D: remove the cell line data, divide the entities by separate figures or different colors, change the order (put the results of the healthy controls at the end of your figure), consider summarizing them
In addition, please be more thorough in your figure legends. Name which exact assay you used, the concentrations (units!), how many replicates you performed, explain short forms in your figures (e.g. Figure 4.A, C, E: An lo PI lo…). This should not be only mentioned in the methods section. Please also explain how you chose the individual concentrations.
Response: We have created a new purged Figure 3 color coded for the different compounds. We have created a new purged Figure 4. We have created a new Figure 5 with a new grouped analysis and moved the former Figure 5 into the supplement Figure S2. For AML cell lines, four independent assays (biological replicates) with four measurements (technical replicates) per dosage were performed. For hematological patient samples, two independent assays with three technical replicates per dosage were performed. This technical information is in the materials and methods section.
Minor concerns
- In line 99, the authors note that they included MDM2 inhibitors in their experiment. Unfortunately, no results are shown or mentioned later. Please include the results or remove the compound from the list.
Response: We have removed the MDM2 inhibitor from the list.
- In Figure 1, the authors compare the response of various cell lines to the STAT5 inhibitor AC-4-130 and the BCL2 inhibitor venetoclax. Is there a reason why you left out the cell lines MOLM-16 and HL-60 in (B)? Also, please change (VC) to (B) in your figure legend to match with “(A)”. Furthermore, it would be interesting if you could interpret your findings on the SMZ-1 cell line, which responded very well to the STAT5 inhibitor, but in contrast, did not respond to venetoclax.
Response: We have added the MOLM-16 and HL-60 response to Venetoclax in Fig.1.
On line 148 we stated that the BCL2 inhibitor venetoclax was more effective in SKM1 and less effective in ML-2 cells in the presence of HS-5 stroma (Fig. S1), indicating a cell and context dependence of BCL2 function. We have added another sentence on line 153. SKM-1 cells response to the BCL2 inhibitor was very low in the absence, but substantial in the presence of HS-5 stroma, indicating BCL2 induction by cytokine receptor signaling in this cell line.
- In Figure 2, the authors show a schematic representation of STAT5 signaling pathways in myeloid cells. Are these cascades already published facts or part of your hypothesis? Please either add your sources or further explain this figure in your discussion. In addition, the name of the FLT3 inhibitor midostaurin (PKC412) is very rarely mentioned in your text, which makes it hard to allocate it to your figure. It would help if you mention the name in Figure 2.
Response: We have added the missing references in the Figure 2 legend and labelled
the FL3 inhibitor midostaurin.
- In Figure 3, the error bars of the DMSO controls show cell viability higher than 100%, which should not be possible for the vehicle control.
Response: We have created a new color coded Figure 3.
- In line 181, please adjust the comma (“in AML cell lines,”, not “in AML, cell lines”).
- The figure legend of Figure 4 is quite confusing due to the similar label for figures A-H and the compounds (“A, C” and “AC”). It would be very helpful if you could find a way to describe one of them differently.
- Please provide more detailed information on the terms and numbers you use in Figure 5.A (“FLT3-TKD 0.57”, “TP53 VAF 92%”, …).
Response: The terms and numbers are stated in the figure legend. For the FLT3 gene
the mutant allele ratio, for the TP53 gene the mutant allele frequency (VAF) are
indicated in parentheses.
- Regarding the in vitro experiments on primary AML cells, it would be interesting to see the coculture experiments with HS-5 stroma cells as well.
Response: This would be interesting, however, there is only a limited amount of AML
patient cells available for testing
- In general, the naming of cell lines and primary cells both as “AML cells” is quite confusing, please consider referring to each differently.
Response: We refer to AML patient cells and AML cell lines. The AML cell lines are AML cells derived from patient samples. As there is the distinction of primary and secondary AML in the patient samples, it would be confusing to refer to the patient samples as primary cells.
Round 2
Reviewer 1 Report
My concerns have been addressed.